

# Identifying myocardial injuries in "normal-appearing" myocardium in pediatric patients with clinically suspected myocarditis using mapping techniques

Haipeng Wang[1], Bin Zhao[2], Huan Yang[1], Tianyi Qian[3], Bo Han[4], Haipeng Jia[5], Jing An[6], Junyu Zhao[7], Ximing Wang[1] and Cuiyan Wang[2]

[1] Department of Radiology, Shandong Provincial Hospital Affiliated to Shandong First Medical University, Ji'nan, China
[2] Shandong Medical Imaging Research Institute Affiliated to Shandong University, Ji'nan, China
[3] Siemens Healthcare, MR Collaborations NE Asia, Beijing, China
[4] Department of Pediatrics, Shandong Provincial Hospital Affiliated to Shandong First Medical University, Ji'nan, China
[5] Department of Radiology, Qilu Hospital of Shandong University, Ji'nan, China
[6] Siemens SSMR, APPL, Beijing, China
[7] Division of Endocrinology, Department of Internal Medicine, Shandong Provincial Qianfoshan Hospital, Ji'nan, China

Corresponding author
Cuiyan Wang,
13869181997@163.com

## ABSTRACT

**Background**. Mapping techniques using cardiac magnetic resonance imaging have significantly improved the diagnostic accuracy for myocarditis with focal myocardial injuries. The aim of our study was to determine whether T1 and T2 mapping techniques could identify diffuse myocardial injuries in "normal-appearing" myocardium in pediatric patients with clinically suspected myocarditis and to evaluate the associations between diffuse myocardial injuries and cardiac function parameters.

**Methods**. Forty-six subjects were included in this study: 20 acute myocarditis patients, 11 subacute/chronic myocarditis patients and 15 control children. T2 values, native T1 values and the extracellular volume (ECV) of "normal-appearing" myocardium were compared among the three groups of patients. Associations between diffuse myocardial injuries and cardiac function parameters were also evaluated.

**Results**. The ECV of "normal-appearing" myocardium was significantly higher in the subacute/chronic myocarditis group than in the control group ($30.1 \pm 0.9$ vs $27.0 \pm 0.6$, $P = 0.004$). No significant differences in T1 and T2 values between the acute myocarditis and control groups were found. In the subacute/chronic myocarditis group, a significant association between ECV and left ventricle ejection fraction was found ($P = 0.03$).

**Conclusions**. Diffuse myocardial injuries are likely to occur in subacute/chronic myocarditis patients with prolonged inflammatory responses. Mapping techniques have great value for the diagnosis and monitoring of myocarditis.

## INTRODUCTION

Myocarditis is a myocardial inflammatory disease associated with various injuries, clinical manifestations and outcomes (*Kindermann et al., 2012*; *Cooper Jr, 2009*; *Zagrosek et al., 2009*). Myocarditis has been identified as a significant cause of sudden death in children (*Blauwet & Cooper, 2010*; *Levine, Klugman & Teach, 2010*). In addition, myocarditis may be an underlying cause of dilated cardiomyopathy (DCM) (up to 40% of DCM cases are caused by myocarditis) (*Heymans et al., 2016*) and may result in death or cardiac transplantation as long as 12 years after diagnosis (*Towbin et al., 2006*).

Cardiac magnetic resonance (CMR) has been an established noninvasive tool for the diagnosis and evaluation of myocarditis (*Luetkens et al., 2014*). Conventional CMR imaging, including T2 weighted imaging, T1 weighted imaging and late gadolinium enhancement (LGE) imaging, is most useful for evaluating focal myocardial injuries by visually comparing the affected area with the normal myocardium (*Friedrich et al., 2009*; *Hamlin et al., 2014*). These combined imaging sequences are an essential part of the "Lake Louise" criteria (2009) *(Friedrich et al., 2009)* and have considerable diagnostic accuracy in myocarditis patients with angina-like symptoms and recent symptom onset (*Luetkens et al., 2016*; *Lurz et al., 2016*; *Radunski et al., 2014*). Diffuse myocardial injuries in myocarditis may present as "normal-appearing" myocardium if it is not compared to normal myocardium. It can also be quantitatively analyzed by normalizing the signals of the myocardium to remote myocardium or skeletal muscles, although the diagnostic accuracy might be affected by abnormal signals of reference muscles (*Friedrich et al., 2009*; *Radunski et al., 2017*).

Currently, T1 and T2 mapping techniques are applied to determine the diagnosis and prognosis of myocarditis. Mapping techniques offer a quantitative assessment of the myocardium by using standardized, reproducible T1 and T2 values and have the potential to identify both focal and diffuse myocardial injuries from myocarditis (*Hamlin et al., 2014*). Extracellular volume (ECV), which is derived from the ratio of pre- and postcontrast T1 values, can measure the fraction of volume occupied by the extracellular space in the myocardium and has become a marker of myocardial tissue remodeling (*Haaf et al., 2016*). In 2018, updated "Lake Louise" criteria (*Ferreira et al., 2018*) were published, and parametric mapping techniques were included in the diagnostic criteria for myocardial inflammation. Compared with that of the original "Lake Louise" criteria, significantly improved diagnostic accuracy has been reported in patients with myocarditis using mapping techniques (*Lurz et al., 2016*; *Radunski et al., 2014*; *Hinojar et al., 2015*). In addition, diffuse myocardial injuries in DCM (*Hong et al., 2015*), myocardial infarctions (*Ugander et al., 2012*; *Chan et al., 2012*) and heart failure (*Mascherbauer et al., 2013*; *Iles et al., 2015*) have been reported by mapping techniques. Myocarditis with focal myocardial injuries has been fully shown using conventional CMR and mapping techniques. However, myocarditis with "normal-appearing" myocardium that might result from diffuse myocardial injuries has received less attention.

In our study, we focused on the "normal-appearing" myocardium—myocardium without focal myocardial edema or necrosis/fibrosis visible on conventional CMR—in

pediatric patients with clinically suspected myocarditis. We attempted to determine whether T1 and T2 mapping techniques could identify diffuse myocardial injuries in pediatric myocarditis patients, and we then evaluated whether there were associations between diffuse myocardial injuries and cardiac function in these patients.

## MATERIALS AND METHODS

In this prospective study, pediatric patients with clinically suspected myocarditis from Feb 2016 to Jan 2018 in our hospital were included. All patients were diagnosed by an experienced pediatrician according to the myocardial diagnostic criteria proposed by the European Society of Cardiology Working Group on Myocardial and Pericardial Diseases (*Caforio et al., 2013*). According to the duration of symptoms from onset to CMR examinations, myocarditis patients were divided into 2 groups: the acute myocarditis (AM) group ($\leq$3 months) and the subacute/chronic myocarditis (CM) group (>3 months). Exclusion criteria were contraindications for CMR, coronary artery diseases, congenital heart diseases, cardiomyopathies, or other medical history of cardiac disease (*Luetkens et al., 2016*). Clinical manifestations, immunological features and electrocardiography (ECG) results were recorded. Children with some mild nonspecific symptoms (such as fatigue, chest congestion) performed CMR examinations to rule out myocarditis. We included 15 control children with normal immunological features, electrocardiography (ECG) and CMR in our study as normal group (NC group). The study was approved by the ethics committee of Shandong Medical Imaging Research Institute, and written informed consent was obtained from the parents of pediatric patients (No. 2016-001).

### CMR imaging protocol

CMR imaging was performed using a MAGNETOM Skyra 3T MR scanner (Siemens Healthcare, Erlangen, Germany) with an 18-channel body matrix coil. All data acquired was retrospectively gated based on the ECG results. Respiratory gating was applied in pediatric patients who could cooperate for a breath-hold during the CMR examinations (usually patients older than 6 years old), and CMR images were acquired at the end-expiratory point. Patients who could not hold their breath (usually younger than 6 years old) were sedated with 10% chloral hydrate solution and examined under free-breathing conditions.

The CMR imaging protocols included T2-weighted imaging, LGE imaging, cine imaging and quantitative image mapping. T2-weighted turbo inversion recovery magnitude (TIRM) sequences were performed in the short-axis (SA) and horizontal long-axis (HLA) orientations (repetition time (TR)/echo time (TE) = 800/44 ms, flip angle (FA) = 180° , field of view (FOV)= 300× 225 mm$^2$, and voxel size = $1.3 \times 1.3 \times 6$ mm$^3$). LGE imaging was performed using phase-sensitive inversion recovery (PSIR) sequences in the HLA and SA orientations 7–10 min after intravenous administration of 0.2 mmol/kg Gd-DTPA (Magnevist, Bayer, Germany). The parameters were as follows: TR/TE= 448/2.0 ms, FOV = 300×350 mm$^2$, matrix = 256×192, and voxel size = $1.4 \times 1.4 \times 6$ mm$^3$. The inversion time (TI) of LGE imaging was determined by using the TI scout. Steady-state free precession (SSFP) cine images were acquired in the HLA and sequential SA orientations

from ventricular base to apex with the following imaging parameters: TR/TE = 39.2/1.4, FA = 80°, FOV = 300× 225 mm$^2$, and voxel size = 1.6× 1.6× 6.0 mm$^3$).

T1 mapping was performed with 3(3)5 modified Look-Locker inversion recovery (MOLLI) sequences in the HLA and three SA orientations (basal, mid and apical ventricular SA planes) (*Lurz et al., 2016*; *Hwang et al., 2014*) before and 15 min after Gd-DTPA administration with the following parameters: TR/TE= 2.4/1.1 ms, FA = 35° , FOV = 300× 225 mm$^2$, acquisition matrix = 256×192 mm$^2$, and voxel size = 1.4 × 1.4 × 8.0 mm$^3$. T2 mapping was acquired using a SSFP sequence with three different T2 preparation times in the HLA and three SA orientations (basal, mid and apical ventricular SA planes). The parameters were as follows: TE = 0 ms, 25 ms, 55 ms; TR = 3 × RR; FA = 50° ; FOV = 300 × 225 mm$^2$; acquisition matrix = 256 × 384 mm$^2$; and voxel size = 0.9 × 0.9 × 8.0 mm$^3$.

## CMR image analysis

All the original image data were processed on the workstation (Siemens Medical Systems). Two experienced CMR radiologists (C.Y.W and H.P.W), who were blinded to patient information, independently analyzed all CMR images.

Left ventricular (LV) cardiac function parameters were evaluated in the cine images. LV endocardial and epicardial contours were drawn manually for each diastolic and systolic frame in the sequential SA cine images, and LV cardiac function parameters, including end-diastolic volume (EDV), end-systolic volume (ESV), left ventricle ejection fraction (LVEF), LV mass and stroke volume (SV), were automatically acquired on the workstation. LV cardiac function parameters were standardized as follows (*Kawel-Boehm et al., 2015*):

Standardized LV cardiac function parameters = LV cardiac function parameters/ body surface area (BSA).

The papillary muscles and trabeculations were included as part of ventricular cavity (*Buechel et al., 2009*).

Myocardial edema and necrosis/fibrosis were defined by visual assessment in the T2-weighted images and LGE images. The presence and location of myocardial edema or fibrosis were independently evaluated by two CMR radiologists according to the 17-segment model proposed by the American Heart Association (AHA) (*Cerqueira et al., 2002*). For the contradictory findings regarding myocardial injuries after independent evaluation, two CMR radiologists would discuss the findings together and reach a consensus. The myocardium without edema or necrosis/fibrosis was defined as "normal-appearing" myocardium, which might include normal myocardium and abnormal myocardium with diffuse myocardial injuries.

T1 and T2 values of "normal-appearing" myocardium in the HLA and SA orientations were measured directly in their T1 and T2 maps. Endocardial and epicardial borders were carefully contoured to exclude artifacts, epicardial fat and blood pools. Then, the T2 values, native T1 values, and postcontrast T1 values of "normal-appearing" myocardium in the HLA and three SA orientations were acquired (Fig. 1). T1 and T2 values in three SA orientations were averaged for data analysis. The extracellular volume (ECV) of the "normal-appearing" myocardium was calculated using native and postcontrast T1 values

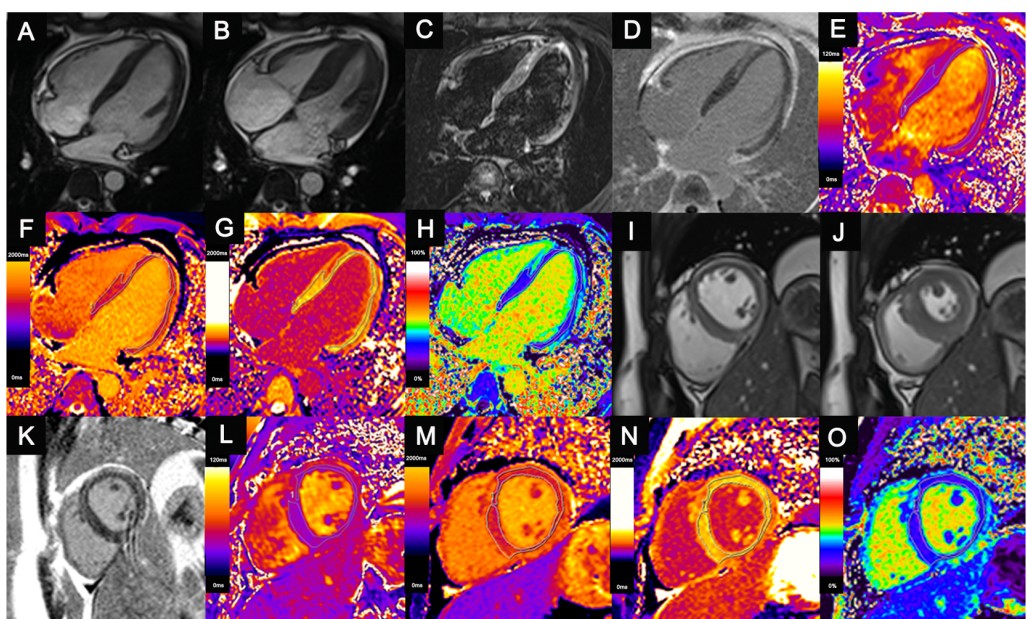

**Figure 1 Comprehensive cardiac magnetic resonance images of a 14-year-old child with acute myocarditis.** He was hospitalized after four days of chest pain. Regional mid-wall myocardial edema of the interventricular septum and epicardial edema of the anterior, lateral, and inferior walls are shown in the end-diastolic (A, I) and end-systolic (B, J) cine images and T2-weighted images (C). Regional myocardial necrosis/fibrosis was also found in the identical location on LGE imaging (D, K). In the T2 maps (E, L), native T1 maps (F, M), post-contrast T1 maps (G, N) and ECV maps (H, O), the dotted line shows the ROI of the "normal-appearing" myocardium excluding visible myocardial edema or necrosis/fibrosis on conventional MRI. T1 values, T2 values and ECVs of "normal-appearing" myocardium were measured as follows: T2 values (HLA) = 37.2 ms, Native T1 values (HLA) = 1,292.4 ms, ECV (HLA) = 25.7%; T2 values (SA) = 37.8 ms, Native T1 values (SA) = 1,297.5 ms, ECV (SA) = 25.9%.LGE, late gadolinium enhancement; ECV, extracellular volume; ROI, region of interest; HLA, horizontal long axis; SA, short axis; ECV, extracellular volume.

of the myocardium and blood pools as well as hematocrit (HCT), as follows (*Luetkens et al., 2014*):

$$ECV(\%) = (1 - HCT) \times (\Delta R1 \text{ of myocardium}/\Delta R1 \text{ of blood pool})$$

$$R1 = 1/T1; \Delta R1 = \text{postcontrast } R1 - \text{native } R1.$$

The native and postcontrast T1 values of the blood pools were also measured directly in the LV cavity, avoiding the papillary muscle. Myocardium with higher T1 and T2 values and ECV was identified as abnormal myocardium with diffuse myocardial injuries.

## Statistical analysis

All analyses were performed using the Statistical Package for the Social Sciences (SPSS, version 19.0) and Empower Stats (http://www.empowerstats.com) software. Categorical data were reported as percentages (%), and continuous data were reported as the mean ±standard deviation (SD) or median (range). The Shapiro–Wilk test was used to assess the normality of the variables using SPSS. Clinical characteristics in the AM, CM and NC

groups were compared using chi-square tests for categorical variables, one-way analysis of variance (ANOVA) for normally distributed continuous variables, and the Kruskal–Wallis test for nonnormally distributed continuous variables. The interaction test and covariate screening were performed to adjust for patient-specific factors using Empower Stats. Linear regression analyses were used to compare T2 values, native T1 values and the ECV of "normal-appearing" myocardium in the HLA and SA orientations among the AM, CM and NC groups. The influence of focal myocardial injuries on T1 and T2 values and the ECV of "normal-appearing" myocardium in myocarditis patients were also evaluated using linear regression analyses. Associations between ECV and LV cardiac function parameters in the AM and CM groups were evaluated using multiple linear regression analyses with adjustment for confounding variables, and odds ratios (ORs) and 95% confidence intervals (CIs) were calculated.

All statistical tests were two-sided, and $P$-values less than 0.05 were considered statistically significant.

## RESULTS

Twenty pediatric patients with acute myocarditis (10 male; median age, 9 years old), 11 patients with subacute/chronic myocarditis (9 male; median age, 6 years old) and 15 control children (9 male; median age, 11 years old) were included in the study.

### Clinical characteristics

The clinical characteristics of all pediatric patients are shown in Table 1. The most common clinical manifestation in the AM group was chest pain/distress, which was present in eight AM patients (40.0%). In the CM group, six patients (54.5%) experienced palpitation. Abnormal cardiac troponin T (cTnT) or brain natriuretic peptide (BNP) levels were observed in nine AM patients (45.0%) and seven CM patients (63.6%). ECG abnormalities were detected in all AM patients and ten CM patients (90.9%). The most common ECG finding in the AM group was ST-T changes (45.0%), while ventricular premature beats (VPBs) were common in the CM group (45.5%).

### CMR findings

The standard LV cardiac function parameters and myocardial tissue characterizations in the AM, CM and NC groups are shown in Table 2. Two AM patients and two CM patients were sedated and underwent CMR examinations under free-breathing conditions, and the remaining patients underwent CMR examinations with respiratory gating. There were no statistically significant differences in standardized EDV, ESV, LV mass, SV and LVEF among the AM, CM and NC groups. Regional myocardial edema or necrosis/fibrosis was found in 7 AM patients (35.0%) and four CM patients (36.4%).

The T1 and T2 values and ECV of "normal-appearing" myocardium in the AM, CM and NC groups are shown in Table 2 and Fig. 2. Compared with that of the NC group, the ECV of the "normal-appearing" myocardium was significantly higher in the CM group after adjusting for BSA, sex, heart rate or HCT (SA: 30.1 ±0.9 vs 27.0 ±0.6, $P = 0.004$). No significant differences in T1 and T2 values were found between the AM and NC groups.

Table 1 **Baseline characteristics of children.** Values are presented as N (%), mean ± SE or median (range).

| Groups | NC ($n = 15$) | AM ($n = 20$) | CM ($n = 11$) | P-values |
|---|---|---|---|---|
| Sex (M, %) | 9 (60.0%) | 10 (50.0%) | 9 (81.8%) | 0.20 |
| Age (y) | 11 (6, 13) | 9 (2, 14) | 6.0 (3, 13) | 0.02[*] |
| HCT (%) | 41.0 ± 0.5 | 39.7 ± 0.8 | 38.4 ± 1.0 | 0.11 |
| HR (/min) | 83.9 ± 3.9 | 91.1 ± 3.5 | 90.3 ± 4.6 | 0.37 |
| Prodrome (%) | – | 11 (55.0%) | – | – |
| CMR intervals | – | 20 days(4, 60) | 8 months(3, 36) | – |
| **Clinical manifestations** | | | | |
| Fatigue | – | 6 (30.0%) | 4 (36.4%) | 0.72 |
| Palpitation | – | 7 (35.0%) | 6 (54.5%) | 0.29 |
| Chest pain | – | 8 (40.0%) | 4 (36.4%) | 0.84 |
| Dyspnea | – | 3 (15.0%) | 3 (27.3%) | 0.42 |
| Heart failure | – | 2 (10.0%) | 0 | 0.18 |
| **Immunological features** | | | | |
| cTNT (pg/ml) | | 3.9(3.0,311.1) | 3.2(3.0,31.4) | 0.33 |
| BNP (pg/ml) | | 79.8(5.0,4606.0) | 166.1(22.0,488.6) | 0.31 |
| **Abnormal ECG** | | | | |
| ST-T changes | – | 9 (45.0%) | 0 | 0.002[*] |
| AVB | – | 8 (40.0%) | 2 (18.2%) | 0.20 |
| IVCB | – | 5 (25.0%) | 2 (18.2%) | 0.66 |
| Tachycardia | – | 3 (15.0%) | 1 (9.1%) | 0.63 |
| APB | – | 1 (5.0%) | 2 (18.2%) | 0.25 |
| VPB | | 5 (25.0%) | 5 (45.5%) | 0.25 |
| Abnormal Q | – | 2 (10.0%) | 1 (9.1%) | 0.93 |
| Sinus bradycardia | – | 2 (10.0%) | 0 | 0.18 |

**Notes.**

[*]*P* values < 0.05.

NC, normal control; AM, acute myocarditis; CM, subacute/chronic myocarditis; HCT, hematocrit; HR, Heart rate; CMR, cardiac magnetic resonance; cTnT, cardiac troponin T; BNP, brain natriuretic peptide; ECG, electrocardiography; AVB, atrioventricular block; IVCB, intra-ventricular conduction block; APB, atrial premature beats; VPB, ventricular premature beat.

To study whether focal myocardial injuries would influence the T1 and T2 values and ECV of "normal-appearing" myocardium, we divided myocarditis patients into another two groups: myocarditis patients with focal myocardial injuries (35.5%) and without focal myocardial injuries (64.5%). No significant differences in T1 or T2 values or ECV between the two groups were found. The T1 and T2 values and ECV of "normal-appearing" myocardium in myocarditis patients with and without focal myocardial injuries are shown in Table 3.

In the CM group, we found negative associations between ECV and LVEF (OR, −0.4; 95% CI, −0.7, −0.1; $P = 0.03$) after adjusting for age, sex, heart rate or HCT. The associations between ECV and LV cardiac function parameters in the CM group are shown in Table 4.

**Table 2  The CMR findings in AM, CM and NC groups.**

| Groups | NC ($n = 15$) | AM ($n = 20$) | CM ($n = 11$) | P-values | | |
|---|---|---|---|---|---|---|
| **Standardized cardiac morphology and function** | | | | | | |
| EDV(mm) | 76.8 ± 4.3 | 74.2 ± 3.4 | 78.3 ± 2.5 | 0.73 | | |
| ESV(mm) | 30.3 ± 2.3 | 30.1 ± 2.0 | 32.4 ± 1.7 | 0.75 | | |
| LVM(g/mm$^2$) | 48.8 ± 2.5 | 47.8 ± 1.9 | 44.6 ± 2.7 | 0.48 | | |
| SV (ml$^{-1}$) | 46.6 ± 2.4 | 44.2 ± 1.6 | 45.9 ± 2.4 | 0.68 | | |
| LVEF(%) | 61.1 ± 1.3 | 60.1 ± 1.1 | 58.6 ± 2.0 | 0.51 | | |
| **Myocardial tissue characterization** | | | | | | |
| T2 (%) | – | 6 (30.0%) | 2 (18.2%) | 0.31 | | |
| LGE (%) | – | 7 (35.0%) | 4 (36.4%) | 0.94 | | |
| **T1 and T2 values of "normal-appearing" myocardium** | | | | $P_1$ | $P_2$ | $P_3$ |
| T2 $_{SA}$, (ms) | 37.2 ± 0.3 | 37.4 ± 0.5 | 37.1 ± 0.2 | 0.54[b] | 0.65[b] | 0.09[b] |
| T1 $_{SA}$ (ms) | 1,297.6 ± 8.0 | 1,328.4 ± 8.0 | 1,320.8 ± 13.0 | 0.09[b] | 0.50[b] | 0.38[b] |
| ECV $_{SA}$(%) | 27.0 ± 0.6 | 28.1 ± 0.5 | 30.1 ± 0.9 | 0.20[*] | 0.004[*] | 0.57[*] |
| T2 $_{HLA}$(ms) | 37.1 ± 0.5 | 36.7 ± 0.4 | 37.2 ± 0.5 | 0.08[b] | 0.90[b] | 0.60[b] |
| T1 $_{HLA}$ (ms) | 1323.3 ± 8.6 | 1336.0 ± 11.5 | 1305.1 ± 8.0 | 0.56[b] | 0.44[b] | 0.05[b] |
| ECV $_{HLA}$ (%) | 28.6 ± 0.5 | 28.5 ± 0.7 | 29.8 ± 1.5 | 0.90[*] | 0.35[*] | 0.58[*] |

**Notes.**

Values are presented as mean ± SE. $P_1$: values comparison between AM and NC; $P_2$: values comparison between CM and NC; $P_3$: values comparison between AM and CM;

[*]P values adjusted for: None

[a]P values adjusted for: BSA and heart rate.

[b]P values adjusted for: BSA, heart rate, hematocrit and sex.

NC, normal control; AM, acute myocarditis; CM, subacute/chronic myocarditis; HLA, horizontal long axis; SA, short axis; ECV, extracellular volume; EDV, end-diastolic volume; ESV, end-systolic volume; LVM, LV mass; SV, stroke volume; LVEF, left ventricle ejection fraction.

# DISCUSSION

In our study, we evaluated the tissue-related changes of "normal-appearing" myocardium in pediatric patients with clinically suspected myocarditis using mapping techniques. We found that ECV could detect diffuse myocardial injuries in "normal-appearing" myocardium in pediatric CM patients, and ECV was associated with LVEF. Therefore, mapping techniques could increase the sensitivity of CMR for monitoring diffuse myocardial injuries in patients with clinically suspected myocarditis.

Mapping technologies could be influenced by numerous factors, including the MR scanner, magnetic field strength, exact sequence used, image acquisition plane, contrast agent dose and patient's physiological differences (*Hamlin et al., 2014*; *Ugander et al., 2012*; *Neilan et al., 2013*; *Sado et al., 2012*; *Liu et al., 2013*). In our study, the scanning protocols of the different mapping technologies were identical, and interaction tests and covariate screenings were performed to adjust for BSA, sex, age, heart rate and HCT of subjects and could minimize the influence of confounding variables.

LGE has been an established noninvasive tool to evaluate focal myocardial necrosis/fibrosis and has shown excellent correlation with pathology (*Luetkens et al., 2014*). In acute "infarct-like" myocarditis, a high sensitivity of LGE has been reported (*Schwab et al., 2016*). However, it is not very sensitive in very mild myocarditis cases, which might have

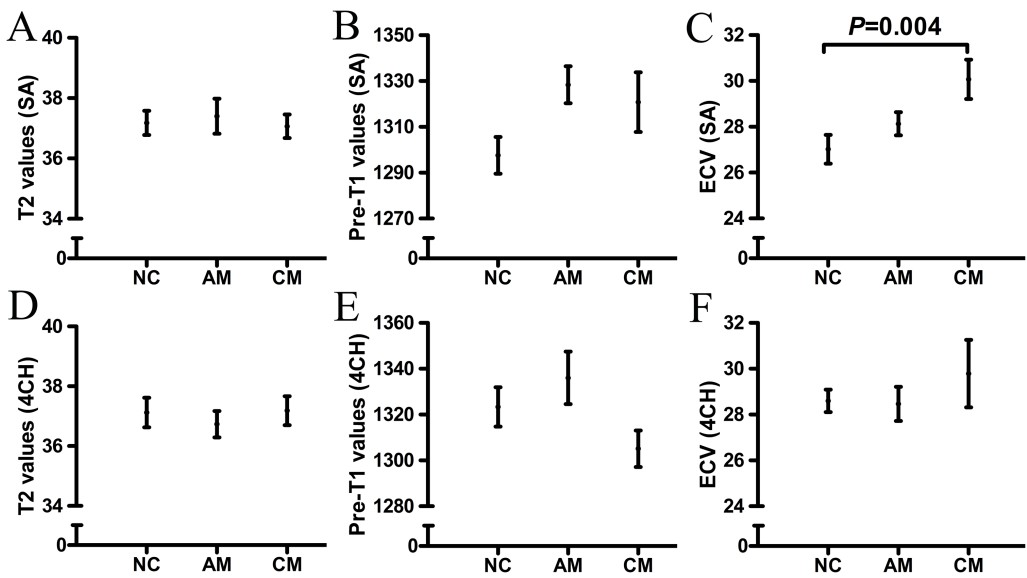

**Figure 2 The T1 and T2 values of "normal-appearing" myocardium in AM, CM and NC groups.** Compared with NC group, the ECV of "normal-appearing" myocardium significantly increased in CM group (C) after adjusted for BSA, sex, heart rate or HCT (SA: 30.1±0.9 VS 27.0±0.6, $P = 0.004$). No significantly statistical differences were found between T1 and T2 values in AM and NC group. AM, acute myocarditis; CM, subacute/chronic myocarditis; NC, normal control; HLA, horizontal long axis; SA, short axis; ECV, extracellular volume.

**Table 3 The T1 and T2 values of "normal-appearing" myocardium in pediatric myocarditis patients with and without focal myocardial injuries on conventional MRI.** Group I: patients with focal myocardial injuries; Group II: patients without focal myocardial injuries.

| | Group I ($n = 11$) | Group II ($n = 20$) | Adjusted ORs (95% CI) | P values |
|---|---|---|---|---|
| T2 $_{SA}$ (ms) | 38.3 ± 0.7 | 36.6 ± 0.3 | − 1.0 (−3.0, 0.9) | 0.30[b] |
| Native T1 $_{SA}$ (ms) | 1338.8 ±12.6 | 1318.5 ±7.8 | −26.5 (−52.5, −0.5) | 0.06[**] |
| ECV $_{SA}$ (%) | 28.7 ±0.9 | 28.9 ±0.6 | 0.0 (−2.1, 2.1) | 0.99[*] |
| T2 $_{HLA}$ (ms) | 37.2 ±0.6 | 36.6 ±0.3 | 0.1 (−1.4, 1.5) | 0.94[d] |
| Native T1 $_{HLA}$ (ms) | 1332.1 ±15.9 | 1323.6 ±8.8 | −3.1 (−51.6, 45.5) | 0.90[d] |
| ECV $_{HLA}$ (%) | 28.4 ±1.2 | 29.1 ±0.7 | 1.2 (−2.5, 4.8) | 0.55[*] |

**Notes.**
Values are presented as mean ± SE.
[*]P values adjusted for: none.
[a]P values adjusted for: BSA, heart rate.
[b]P values adjusted for: BSA, heart rate and hematocrit.
[**]P values adjusted for: BSA.
[c]P values adjusted for: BSA, heart rate and sex.
[d]P values adjusted for: BSA, heart rate, hematocrit and sex.
ORs, odds ratios; CI, confidence interval; HLA, horizontal long axis; SA, short axis; ECV, extracellular volume; BSA, body surface area.

diffuse myocardial tissue-related changes. In contrast to LGE, ECV is well suited to measure focal and diffuse myocardial fibrosis and exhibits the best agreement with histological measures of the collagen volume fraction. ECV has been shown to be reproducible, predict

**Table 4  The associations between ECV and cardiac function in subacute/chronic myocarditis patients.**

|  | Adjusted ORs (95% CI) | P Value |
|---|---|---|
| EDV | −0.1 (−0.2, 0.1) | 0.30[*] |
| ESV | −0.1 (−0.4, 0.3) | 0.78[*] |
| SV | −0.2 (−0.3, 0.0) | 0.15[*] |
| LVM | −0.1 (−0.2, 0.1) | 0.44[a] |
| LVEF | −0.4 (−0.7, −0.1) | 0.03[*] |

**Notes.**

[*]P values adjusted for: age, sex, heart rate and hematocrit.

[a]P values adjusted for: age and hematocrit.

ECV, extracellular volume; LV, left ventricular; CM, subacute/chronic myocarditis; ORs, odds ratios; CI, confidence interval; EDV, end-diastolic volume; ESV, end-systolic volume; LVM, left ventricular mass; SV, stroke volume; LVEF, left ventricle ejection fraction.

outcomes and provide "added prognostic value" in myocardial disease (*Ferreira et al., 2018*; *Messroghli et al., 2017*). For myocarditis, ECV has been included in the updated "Lake Louise" criteria (*Ferreira et al., 2018*), which could certainly greatly improve the diagnostic sensitivity for myocarditis over that of the original "Lake Louise" criteria, especially in myocarditis with diffuse myocardial injuries.

In our study, we found that the ECV of "normal-appearing" myocardium was significantly higher in pediatric CM patients than in NC patients, which indicated diffuse myocardial injuries. Myocarditis has been identified as an underlying cause of DCM, and up to 40% of DCM cases are caused by myocarditis (*Heymans et al., 2016*). The availability of murine models of myocarditis has facilitated much of our understanding of the pathogenesis of myocarditis-DCM (*Sagar, Liu & Cooper Jr, 2012*). During the progression of myocarditis, inflammatory cells embedded in the interstitial matrix contribute to the inflammatory response and cardiac remodeling. The expansive interstitial matrix could be measured by ECV (*Haaf et al., 2016*). In our study, diffuse myocardial injuries were more likely to occur in CM patients than in AM patients. Abnormal cTnT or BNP was observed in 63.6% of CM patients during CMR examination, which was higher than that in AM patients (45.0%). We hypothesized that expansive interstitial matrix deposition was likely to occur in CM patients with prolonged and recurrent inflammatory responses. ECV might be a marker of myocarditis leading to DCM. The values of ECV related to the outcomes of CM patients have been followed up and will be discussed in future studies. ECV quantification of interstitial expansion remains a powerful tool to investigate diffuse myocardial injuries.

In our study, native T1 values of the "normal-appearing" myocardium were not significantly higher in AM and CM patients than in NC patients, which was inconsistent with recent data by *Radunski et al. (2017)*. Radunski UK et al. found that native T1 values in the "normal-appearing" myocardium of AM patients were significantly higher than the reference values from the myocardium of healthy volunteers. This discrepancy could be explained by the fact that all the AM patients who Radunski UK included had typical focal myocardial LGE findings, while the inflammatory response in our study was mild. Despite this, a higher ECV of "normal-appearing" myocardium was observed in the CM group. Native T1 values perform as composite indicators of both intracellular

and extracellular compartments (*Burt et al., 2014*) and, therefore, can be less sensitive to increased extracellular space or more sensitive to other characteristics of the tissue (such as increased iron content, fatty deposition, and edema) (*Burt et al., 2014*). ECV is derived from the ratio of T1 signal values and simply quantifies the interstitial presence of gadolinium relative to plasma (*Haaf et al., 2016*). ECV represents a physiological parameter, and its values are therefore reproducible. Therefore, ECV could reflect diffuse myocardial injuries with more sensitivity than native T1 values.

In our study, T2 values in the "normal-appearing" myocardium of AM patients was not significantly higher than those in the NC group, which was in agreement with the findings of *Radunski et al. (2017)*. In addition to the disadvantages of T2 mapping due to unstable myocardial edema (*Radunski et al., 2017*), we also reasoned that mild myocardial inflammation in our study would have influenced the results.

In our study, we found associations between ECV and LVEF in CM patients, which have also been reported in patients with diabetic cardiomyopathy, myocardial infarction, hypertrophic cardiomyopathy and heart failure (*Hong et al., 2015*; *Ugander et al., 2012*; *Chan et al., 2012*; *Mascherbauer et al., 2013*; *Iles et al., 2015*). The pathophysiology of myocarditis in murine models suggests that a persistent inflammatory response in the chronic phase of myocarditis leads to ventricular remodeling, which is characterized by myocyte hypertrophy, myocyte apoptosis, contractile dysfunction and extracellular matrix volume expansion (*Kindermann et al., 2012*; *Cooper Jr, 2009*; *Hinojar et al., 2015*; *Elamm, Fairweather & Cooper, 2012*). ECV has become a marker of myocardial tissue remodeling (*Haaf et al., 2016*), and it could predict outcomes and provide "added prognostic value" in myocardial disease (*Ferreira et al., 2018*; *Messroghli et al., 2017*). Early data indicate that ECV appears to be as prognostically important as LVEF (*Eitel et al., 2015*; *Sanguineti et al., 2015*), which underestimates the biological importance of the interstitium. In our study, CM patients had increased ECV and normal LVEF. ECV is comparable to LVEF as a marker to evaluate myocardial injuries.

This study has several potential limitations. First, the myocarditis patients involved in our study were diagnosed according to the diagnostic criteria for clinically suspected myocarditis proposed by the ESC Working Group on Myocardial and Pericardial Diseases (*Caforio et al., 2013*). Endomyocardial biopsy should be the gold standard for the definitive diagnosis of myocarditis. However, it might be unrealistic to perform biopsies in most pediatric myocarditis patients. In our study, 85.0% AM patients and 72.7% CM patients fulfilled more than 3 criteria for clinically suspected myocarditis, which increased the strength of the suspicion for myocarditis. Second, the severity of myocardial inflammation was less severe than that reported in other studies. We did not acquire significant results with native T1 mapping and T2 mapping. The diagnostic efficacy for myocarditis according to the "Lake Louise" criteria was low. Third, the intervals from onset to CMR examinations varied based on the patient's condition, which might have influenced the CMR findings. Fourth, the number of pediatric myocarditis patients included in the study was limited. Analyses of larger populations should be performed in the future.

## CONCLUSION

In conclusion, diffuse myocardial injuries are likely to occur in CM patients with prolonged inflammatory responses. Mapping techniques have great value for the diagnosis and monitoring of myocarditis.

## ACKNOWLEDGEMENTS

We would like to acknowledge the participation of the study patients and their families. We also wish to thank the Department of Pediatrics for providing partial clinical data of the pediatric patients.

### Funding

This work was supported by the Shandong Provincial Key Research and Development Program (No. 2019GSF108202). The funders had no role in study design, data collection and analysis, decision to publish, or preparation of the manuscript.

### Grant Disclosures

The following grant information was disclosed by the authors:
Shandong Provincial Key Research and Development Program: 2019GSF108202.

### Competing Interests

Tianyi Qian is employed by Siemens Healthcare and Jing An is employed by Siemens SSMR. All other authors declare that they have no competing interests.

### Author Contributions

- Haipeng Wang conceived and designed the experiments, performed the experiments, analyzed the data, prepared figures and/or tables, and approved the final draft.
- Bin Zhao, Bo Han and Cuiyan Wang conceived and designed the experiments, authored or reviewed drafts of the paper, and approved the final draft.
- Huan Yang performed the experiments, analyzed the data, prepared figures and/or tables, and approved the final draft.
- Tianyi Qian, Haipeng Jia and Jing An performed the experiments, authored or reviewed drafts of the paper, and approved the final draft.
- Junyu Zhao analyzed the data, prepared figures and/or tables, and approved the final draft.
- Ximing Wang analyzed the data, authored or reviewed drafts of the paper, and approved the final draft.

### Human Ethics

The following information was supplied relating to ethical approvals (i.e., approving body and any reference numbers):

The study was approved by the ethics committee of Shandong Medical Imaging Research Institute (No.2016-001) and written informed consent was obtained from the parents of pediatric patients.

## Data Availability

The raw measurements are available in the Supplementary File.

## Supplemental Information

Supplemental information for this article can be found online at http://dx.doi.org/10.7717/peerj.10252#supplemental-information.

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
