# Peer review of "Identifying myocardial injuries in “normal-appearing” myocardium in pediatric patients with clinically suspected myocarditis using mapping techniques"

_PeerJ, doi:10.7717/peerj.10252_

## Round 0.1 · original submission · Major Revisions

The concerns that the reviewers raised should be carefully addressed.

Reviewer 1 ·

Basic reporting

This is an interesting study which focused on "normal-appearing" myocardium in pediatric patients with clinical suspected myocarditis. The authors try to identify myocardial injuries of "normal-appearing" myocardium using T1 mapping and T2mapping techniques. They compared the T2 values, native T1 values and ECV of "normal-appearing" myocardium among acute myocarditis patients, subacute/chronic myocarditis patients and healthy children. The results showed that diffuse myocardial injuries are prone to occur in subacute/chronic myocarditis with prolonged inflammatory response. The study affirmed the value on the diagnosis and monitoring of myocarditis of mapping techniques.

I have the following comments for the authors to embellish:

1. In the “Methodology” (line 36), “among” maybe better than “between”.
“T2 values, native T1 values and extracellular volume (ECV) 36 of “normal-appearing” myocardium between three groups were compared. ”

2. In the “Discussion” (line 279), “characteristic” maybe appropriate than “characterization”.
"However, it is not very sensitive in very mild cases, which might perform diffuse myocardial tissue characterization changes."

3. The sentence in line 297 maybe unnecessary.
"Diffuse myocardial injuries in DCM have been reported by mapping techniques."

Experimental design

The study design is sound for purpose and the results are well presented.

CMR imaging protocol: How many SA slices were performed during mapping sequences? And how to decide the position, for example “basal, mid and apical ventricular SA planes”.

Validity of the findings

Please provide the p values between AM and CM in table 2.

Additional comments

no comment

Reviewer 2 ·

Basic reporting

This manuscript was well-written with clear and unambiguous professional English. Introduction part and references gave adequate background information, and demonstrated the novelty of this study. Tables and figures were well-prepared.

Experimental design

This manuscript addressed an interesting topic of diffused injury in “normal-appearing” myocardium of patients with myocarditis, demonstrating that subacute/chronic myocarditis patients may show higher ECV even in non-LGE non-edema myocardium, and higher ECV may be correlated to impaired LVEF. The authors employed T1/T2 mapping technique, and provided plenty of details of this study in Methods part.
Line 165-166 states that “The papillary muscles and trabeculations were included as part of LV mass”, while reference 26 says “For calculation of global ventricular volumes, mass and function the papillary muscles were included in the ventricular cavity”. I think it might be a typo.
It would be better if the authors gave more information about the criteria of included region (normal-appearing myocardium)/excluded region (focal myocardial edema or necrosis/fibrosis). For example, (line 168-170,177-178) Were focal myocardial edema in T2WI and necrosis/fibrosis in LGE defined by visual assessment?

Validity of the findings

The authors described results of their study comprehensively and provided robust raw dataset of studied CMR parameters, which supported the conclusion of this study.

Additional comments

no comment

·

Basic reporting

no comment

Experimental design

no comment

Validity of the findings

no comment

Additional comments

In this study, the authors analyzed diffuse myocardial injuries of “normal-appearing” myocardium in pediatric patients with clinical suspected myocarditis using T1 and T2 mapping techniques and evaluated the associations between diffuse myocardial injuries and cardiac function parameters. The authors found that the extracellular volume of “normal-appearing” myocardium significantly increased in subacute/chronic myocarditis compared with controls and it was associated with LVEF.
The authors need to address the following:
1. There are too many language questions in this article. I suggest the manuscript be edited by a native speaker, or correct the grammar issues cautiously.
2. Abstract: Line 33: “Methodology” or “Methods” ?
3. Line 64-66: Diffuse, as opposed to focal, myocardial injuries in myocarditis may perform as “normal-appearing” myocardium without comparison of normal myocardium. This sentence is confusing.
4. Line 108: 15control
5.How were 15 control children included in this study? Please add the corresponding inclusion and exclusion criteria.
6.The authors focused on the “normal-appearing” myocardium, which was defined as no focal myocardial edema or necrosis/fibrosis in the T2-weighted and LGE images in this study (line 176-178). Did it also assess normal myocardium? How to distinguish between normal myocardium and diffuse myocardial injuries, I think it should be explained in the Materials and methods.
7. Sample size was small and all the subjects were male (line 216-219), the results were limited and it might not be sufficient to support the conclusion.
8. Line 275-277: LGE has become the “golden standard” to evaluate focal myocardial necrosis/fibrosis or other forms of irreversible injury infiltration in myocardial diseases. I think this sentence is not accurate. Pathological examination is usually the gold standard to evaluate the degree of myocardial necrosis and injury, but it is difficult to carry out in living people.
9. The duration of symptoms to CMR timing range is broad in CM group. Does this affect results?
10. Table 4 shows the associations between ECV and cardiac function in subacute/chronic myocarditis patients. Is there the data that shows the associations between ECV and cardiac function in all groups?

---

## Round 0.2 · accepted · Accept

All reviewers were satisfied with the revision of this manuscript.

Reviewer 1 ·

Basic reporting

No comment

Experimental design

No comment

Validity of the findings

No comment

Reviewer 2 ·

Basic reporting

no comment

Experimental design

no comment

Validity of the findings

no comment

Additional comments

The authors have addressed my concerns. No further comments.

·

Basic reporting

no comment

Experimental design

no comment

Validity of the findings

no comment

Additional comments

no comment